# Temporal and Spatial Differentiation and Driving Factors of China's Agricultural Eco-Efficiency Considering Agricultural Carbon Sinks

**Shilin Li** [1,2]**, Zhiyuan Zhu** [1,2]**, Zhenzhong Dai** [1,2]**, Jiajia Duan** [1,2]**, Danmeng Wang** [1,2] **and Yongzhong Feng** [1,2,*]

1 College of Agronomy, Northwest A&F University, Yangling 712100, China
2 The Research Center of Recycle Agricultural Engineering and Technology of Shaanxi Province, Yangling 712100, China
* Correspondence: fengyz@nwsuaf.edu.cn; Tel.: +86-029-87080736; Fax: +86-029-8709-2265

**Abstract:** Climate change, greenhouse gas emissions, and food security have put forward higher requirements for sustainable agricultural development. Agricultural ecological efficiency (AEE) is an important indicator to evaluate the sustainable development of agriculture. Low carbon agriculture promotes sustainable agricultural development. Agricultural carbon sinks are an important output of agricultural production, but they have not been fully reflected in the current research on agricultural ecological efficiency. In this study, agricultural carbon sinks are considered as one of the expected outputs of AEE. The data envelopment method was used to measure the AEE of 31 provincial-level administrative regions in China from 2000 to 2019, and the AEE of China was compared with and without carbon sinks. The Gaussian kernel function was used to estimate the time evolution of regional differences in AEE. A geodetector model was used to detect the drivers of spatial differentiation of AEE in China. The results showed that considering agricultural carbon sinks as one of the expected measurement outputs brings the estimated AEE closer to reality. From 2000 to 2019, China's AEE showed an upward trend, and the efficiency value increased from 0.48 to 0.95, an increase of 97.92%. The spatial distribution pattern of AEE in China was Northeast > West > Central > East, with obvious differences among provinces. The industrialization level, urban–rural gap, agricultural economic level, agricultural disaster rate, and urbanization level were the leading driving forces for the spatial differentiation of AEE in China. The research will help to reveal the dynamic characteristics, spatial differentiation characteristics, and driving factors of China's agricultural ecological efficiency, and provide a scientific reference for the realization of sustainable agricultural development and high-quality development.

**Keywords:** agricultural carbon sink; eco-efficiency; geodetector; factor

## 1. Introduction

With global climate change, natural disasters are occurring more and more frequently, and greenhouse gas emissions and food security issues have begun receiving extensive attention [1,2]. The "2022 World Food Security and Nutrition Status Report" showed that hunger affects 100 million people worldwide, with approximately 29.3% of the world's population being moderately or severely food insecure. In the face of climate change and the depletion of natural resources, the global situations of hunger and food insecurity are becoming increasingly severe. Therefore, resource coordination and environmentally friendly and sustainable agricultural development are particularly important for addressing these issues. Meeting the growing demand for food production, efficiently utilizing existing inputs and outputs, producing more products with fewer resources, and avoiding losses and waste, all while reducing the adverse effects of air pollution and greenhouse gas emissions from agricultural production, are the main challenges and opportunities for the sustainable development of the agricultural industry [3].

With the development of the economy and society, China's agricultural economy has been developing rapidly at a high cost. A series of problems caused by the excessive exploitation of agricultural resources and excessive use of inputs such as chemical fertilizers and pesticides have become increasingly prominent. The sustainability of agricultural development is facing major challenges. Sustainable agricultural production emphasizes ensuring economic benefits while protecting the local ecological environment, which requires the coordination of economic development and environmental protection [4,5]. The concept of eco-efficiency, or the ratio of added value to increased environmental impact [6], was proposed by Schaltergger in 1990 [7], and was subsequently promoted by organizations such as the World Business Council (WBCSD) and the Organization for Economic Cooperation and Development (OECD). Since its introduction, eco-efficiency has gradually become an important tool for measuring sustainable development. Eco-efficiency emphasizes the unification of economic and environmental benefits, or in other words, minimizing resource consumption and environmental pressure while maximizing output [8]. Agricultural ecological efficiency (AEE) is the expansion of ecological efficiency to the agricultural field, specifically referring to obtaining higher agricultural output with as little input as possible, while simultaneously reducing resource waste and environmental pollution as much as possible. Ultimately, AEE achieves agricultural economic benefits as well as environmental protection through the coordinated and unified development of these benefits. The scientific evaluation of the temporal and spatial changes in AEE and the exploration of its driving factors are of great significance for improving AEE and promoting sustainable agricultural development.

At present, the research on AEE is relatively abundant and extensive, mainly at the levels of agricultural production at the national, urban, and regional scales [9–18]. In terms of research content, it mainly focuses on efficiency measurement time series evolution and influencing factors [19–25]. In China, most measurements of AEE are based on only 30 provinces across the country, excluding Tibet; thus, they do not fully reflect China's AEE.

Agricultural carbon sinks describe the carbon absorbed by agricultural crops [26]. Agricultural activities that purify the atmosphere of carbon dioxide are regarded as agro-ecological welfare. In terms of measurement indicators, existing research lacks the consideration of agricultural carbon sinks. Furthermore, there have been relatively few studies of AEE at the regional level, and the interactions between its overall driving factors. It should also be noted that the existing research focuses on the planting industry. From the perspective of China's agricultural output value, the current value from the planting industry accounts for approximately 50% of the total agricultural output value, indicating that the traditional agricultural production model based on planting has been diversified [27,28]. In addition, the proportion of non-plantation pollutant emissions in agricultural pollutants is close to 75%, which means that if the ecological efficiency of the planting industry is used to represent China's AEE, the estimated results will indicate a gap in agricultural production [14].

In terms of measuring AEE, the main methods are the ratio method, life cycle assessment, stochastic frontier analysis (SFA), energy analysis, and data envelopment analysis [29–32]. In most cases, analytical methods such as data envelopment analysis (DEA) are the most important methods for obtaining highly correlated results [33]. SFA is a parameterization method that is generally only suitable for single-output and multiple-input production [34], which biases the calculated efficiency value; on this account, Tone proposed the SBM model of undesired output in 2001 [35]. However, similar to the traditional DEA model, the SBM model cannot further distinguish DMUs with an efficiency of one. In 2002, Tone [36] built a super-efficient SBM model that can effectively avoid the shortcomings of other methods such as single-index, multi-index, principal component, and decoupling analyses. Undesired outputs such as environmental pollution are incorporated into the objective function to distinguish the differences between the effective DMUs, and the calculation results are relatively more accurate than those of other methods.

Based on the above, in order to achieve agricultural emission reduction and encourage resource-saving and environmentally friendly agricultural production, the purposes of the study are: (1) to build an AEE evaluation model including agricultural carbon sink, (2) to explore the temporal and spatial evolution characteristics of AEE in China's regions and provinces from 2000 to 2019, (3) to identify the driving factors of AEE change in different regions and provinces.

## 2. Materials and Methods

### 2.1. Data Sources

The study area includes 31 provinces in mainland China. Data on the number of people in the primary industry, agricultural sown area, chemical fertilizers, pesticides, agricultural plastic films, and the number of students in school were obtained from the official website of the National Bureau of Statistics of China. Agricultural production and socio-economic data were from the China Rural Statistical Data, the China Rural Statistical Yearbook, and the statistical yearbooks and bulletins of various provinces, autonomous regions, and municipalities. Interpolation methods were used to fill in the missing data. Precipitation data were obtained from the "Annual Value Dataset of Surface Climate Data in China" of the China Meteorological Data Network. Referring to the Hu Jianglin [37] Barnes method, the Inverse Distance Weighted (IDW) method was used to interpolate the grid point data, and then the regional average was calculated. Average annual precipitation data were determined for each province. Basic geographic information data (national and provincial borders) were obtained from the national 1:4 million basic geographic database (webmap.cn) of the National Geographic Information Center. Data on the cultivated land area were obtained from the "China Statistical Yearbook" and the RESSET database. All data accessed from 1 February to 26 March 2022.

### 2.2. Methods

#### 2.2.1. Super-SBM Model with Unexpected Outputs

The super-efficient SBM model based on undesired output mentioned above mainly considers factors such as environmental pollution. The undesired output is included in the objective function, and then the difference between the effective DMUs is distinguished. The specific model is constructed as follows:

$$m \text{ in } \rho = \frac{\frac{1}{m} \sum_{i=1}^{m} \left( \frac{\overline{x}}{x_{ik}} \right)}{\frac{1}{r_1+r_2} \left( \sum_{s=1}^{r_1} \frac{\overline{y^d}}{y^d_{sk}} + \sum_{q=1}^{r_2} \frac{\overline{y^u}}{y^u_{uk}} \right)}$$

$$\overline{x} \geq \sum_{j=1,\neq k}^{n} x_{ij}\lambda_j; \overline{y^d} \leq \sum_{j=1,\neq k}^{n} y^d_{sj}\lambda_j; \overline{y^d} \geq \sum_{j=1,\neq k}^{n} y^d_{qj}\lambda_j;$$

$$\overline{x} \geq x_k; \overline{y^d} \leq y^d_{sj}; \overline{y^u} \leq y^u_{uk};$$

$$\lambda_j \geq 0, i = 1, 2, \ldots, m; j = 1, 2, \ldots n; j \neq 0;$$

$$s = 1, 2, \ldots r_1; q = 1, 2, \ldots, r_2.$$

In the formula, $n$ is the number of DMUs; $j$ is the $j$th DMU; $k$ is the $k$th DMU calculated by the current efficiency; $x$ is the input index; $m$ is the number of input indexes; $i$ is the $i$th input index; $y^d$ and $y^u$ are the expected output and undesired output indicators, respectively; $r_1$ and $r_2$ are the quantities of expected output and undesired output indicators, respectively; $s$ and $q$ represent the $s$th and $q$th expected output and undesired output indicators, respectively; $\rho$ is the AEE value; and $\lambda$ represents the weight.

#### 2.2.2. Kernel Density Function

The kernel density function is a non-parametric estimation method based on the kernel function [38]. It is used to estimate the probability density of random variables by

smoothing to determine the distribution of random variables. When the probability density function of a set of random variables is *f(x)*, the expression of the model is:

$$f(x) = \frac{1}{nh} \sum_{i=1}^{n} K\left(\frac{x - x_i}{h}\right)$$

where *n* is the number of observations, *K(·)* is the kernel density function, and *h* is the bandwidth. The optimal *h* should be selected to minimize the integral mean square error. There are various kernel functions, including the Gaussian kernel and quadratic kernel. Compared with other kernel functions, the Gaussian kernel function has the best estimation effect. Therefore, in this study, the Gaussian kernel function was used to estimate the AEE kernel density curve to analyze the time evolution of regional differences in AEE [39].

### 2.2.3. Geodetector

The geodetector is a statistical tool for studying the spatial heterogeneity of geographical phenomena and revealing their driving factors [40]. The basic assumption of geographic detectors is that several sub-regions are distinguished. If the sum of the variances of the sub-regions is less than the total regional variance, there is spatial heterogeneity; if the spatial distribution of two variables tends to be consistent, there is a statistical correlation between the two variables [41,42]. The functions of geodetectors include factor detection and risk detection. There are four sub-detectors of interactive detection and ecological detection, among which the most commonly used are differentiation, factor detection, and interactive detection. That is, the *q* value and *p* value are calculated to reveal the spatial heterogeneity of geographical elements and explore their influencing factors. This study mainly used factor probing and interaction probing.

1. Factor detection. The factor detector was used to calculate the *q* value of each factor, which was used to quantitatively analyze the spatial differentiation of AEE and to detect the extent to which a factor explained the spatial differentiation. The formula is:

$$q = 1 - \frac{\sum_{h=1}^{m} N_h \sigma_h^2}{N \sigma^2}$$

   where $h = 1, 2, \ldots ; m$ is the stratification or partition of the independent variable *X* and the dependent variable *Y*; $N_h$ and *N* are the number of units in layer *h* and the whole area, respectively; $\sigma_h^2$ and $\sigma^2$ are the variances of the *Y* values of layer *h* and the whole area, respectively; and *q* is a measurement of the explanatory power, with a range of 0 to 1. The larger the *q* value, the stronger the explanatory power of the independent variable *X* to the dependent variable *Y*, and vice versa.

2. Interactive detection was used to identify the interaction between different independent variables, that is, to determine whether the interaction of influencing factors will enhance or weaken the explanatory power of AEE, or whether the influencing factors act independently, the detection calculation formula were from Wang [40].

### 2.2.4. Indicator Selection

1. Explained variable

   Based on the concept of agricultural carbon sinks, the previous research, and the actual agricultural production in China, this study selected factors such as labor, land, irrigation, agricultural machinery, chemical fertilizers, pesticides, and agricultural film as input variables, and selected the total output value of agriculture, agriculture carbon sinks, and agriculture carbon emissions as output variables. The total output value of agriculture and agricultural carbon sinks were the expected outputs, and agricultural carbon emissions were the undesired outputs. Based on this, a measurement index system of China's AEE was constructed (Table 1). There are three main sources of agricultural carbon emissions. One is the greenhouse gas generated by livestock breeding, which mainly includes $CH_4$

emissions caused by enteric fermentation as well as $CH_4$ and $N_2O$ emissions caused by manure management and treatment. Therefore, this study selected seven categories of livestock, including cattle, horses, and donkeys, to calculate their carbon emissions. Another source of emissions is the greenhouse gas generated by chemical fertilizers, pesticides, agricultural film, agricultural diesel, agricultural irrigation, and agricultural machinery during the agricultural production process. The final source is the $CH_4$ emissions from rice fields. The carbon emissions from these latter two sources were also calculated. The above-mentioned emission sources were multiplied by the corresponding emission coefficients to obtain estimations. The calculation formula and correlation coefficients were obtained from existing studies by Min et al. [43–46]. The agricultural carbon sink was calculated based on the total amount of carbon dioxide absorbed by different crops during photosynthesis. The calculation formula and correlation coefficient were obtained from existing studies by Han et al. [47–50].

**Table 1.** Index system of AEE measurement.

| Main Variable | Type | Specific Indicators | Variable Description | Data Sources |
|---|---|---|---|---|
| | Labor consumption | Labor input | Number of people in the primary industry/$10^4$ | stats.gov.cn |
| | Material consumption | Land input | Crop sown area/$10^3$ $hm^2$ | stats.gov.cn |
| | | Water input | Effective irrigation area/$10^3$ $hm^2$ | stats.gov.cn |
| Unput indicator | | Agricultural machinery input | Agricultural mechanization/$10^4$ kw | China Rural Statistical Yearbook |
| | Environmental cost | Fertilizer input | Fertilizer application rate/$10^4$ t | stats.gov.cn |
| | | Pesticide input | Pesticide usage/$10^4$ t | stats.gov.cn |
| | | Agricultural film input | Amount of plastic film used/$10^4$ t | stats.gov.cn |
| | Expected output | Agricultural output value | Agricultural output value/$10^8$ ¥ | stats.gov.cn |
| Output indicator | | Carbon sink | Agricultural production carbon sink/$10^4$ t | by Han et al. [46–49] |
| | Undesired output | Carbon emission | Total agricultural carbon emissions/$10^4$ t | by Min et al. [42–45] |

2. Explanatory variables

AEE is a systematic problem that is not only affected by internal resource conditions such as the endowment of agricultural production itself, but also by external conditions such as social and economic development, the natural ecological environment, and policy support. In view of this, this study selected independent variables from four aspects of agricultural resource endowment, agricultural economic development, social environment, and policy support, including eleven specific indicators, to construct a driving factor indicator system for AEE (Table 2).

**Table 2.** Driving factors and classification of AEE.

| Type of Representation | Driving Factors | | Code | Data Sources |
|---|---|---|---|---|
| | Driver | Variable Description and Calculation | | |
| | Per capita arable land | Area of arable land/resident population at the end of the year | X1 | resset.com |
| Natural Resources | Agricultural disaster rate | Affected area/total sown area | X2 | stats.gov.cn |
| | precipitation | Average annual precipitation | X3 | data.cma.cn |

**Table 2.** *Cont.*

| Type of Representation | Driving Factors | | Code | Data Sources |
| --- | --- | --- | --- | --- |
| | **Driver** | **Variable Description and Calculation** | | |
| Agricultural Development | Agricultural economic level | Gross Agricultural Output/Number of Permanent Residents | X4 | stats.gov.cn |
| | Industrial structure | Gross agricultural output value/gross output value of agriculture, forestry, animal husbandry and fishery | X5 | stats.gov.cn |
| | Degree of agricultural mechanization | Total power of agricultural machinery/total sown area of crops | X6 | stats.gov.cn |
| Social Environment | Urbanization level | Urban Population/Total Population | X7 | stats.gov.cn |
| | Level of industrialization | Industrial value added/Gross regional product | X8 | stats.gov.cn |
| | urban–rural gap | Per capita disposable income of urban residents/per capita disposable income of rural residents | X9 | stats.gov.cn |
| | Years of education per capita in rural areas | (Number of primary school students * 6 + Number of junior high school students * 9 + Number of people above high school * 16) Total number of people | X10 | stats.gov.cn, China Rural Statistical Yearbook |
| Policy Support | The level of financial support for agriculture | Fiscal expenditure on agriculture, forestry, and water/financial general public budget expenditure | X11 | China Statistical Yearbook, Finance Yearbook Of China |

## 3. Results

### 3.1. Measurement Analysis of AEE with and without Carbon Sinks

Agricultural carbon sinks are ecological benefits that accompany agricultural production practices. In this study, the super-efficient SBM model was used to measure the AEE of each province in China. Figure 1 reports the AEE of China with and without agricultural carbon sinks. When carbon sinks were considered, the AEE was significantly higher than that when carbon sinks were not considered.

As can be seen from Table 3, when carbon sinks were not considered, only five provinces had AEEs greater than 0.5. According to the division standard of the four major economic divisions of the National Bureau of Statistics, the eastern region (including Beijing, Tianjin, Hebei, Jiangsu, Zhejiang, Shanghai, Fujian, Guangdong, Shandong, Hainan), the central region (including such as Shanxi, Hunan, Hubei, Henan, Jiangxi, Anhui), the western region (including Tibet, Inner Mongolia, Guangxi, Chongqing, Sichuan, Guizhou, Yunnan, Shaanxi, Gansu, Qinghai, Ningxia, Xinjiang), and the northeast region (including Liaoning, Jilin, Heilongjiang), had average values of 0.35, 0.36, 0.38, and 0.27, respectively. Thus, the order of the four regions were West > Central > Eastern > Northeast.

When considering agricultural carbon sinks, the AEE of 22 provinces was greater than 0.5, indicating that AEE was greatly improved when agricultural carbon sinks were considered. Under this scenario, the average AEE in the northeast region has increased the most, with an AEE growth rate of 168.22%, followed by the central region with 75.23%, the western region with 64.64%, and the eastern region with 56.23%. The AEE performance was in the order of Northeast > West > Central > East. This may be because better agricultural carbon sinks effectively raise the AEE in these places. The western plateau, underdeveloped regions, and northeastern regions are sparsely populated, rich in forest resources, and of high ecological importance, thus providing better agricultural carbon sinks.

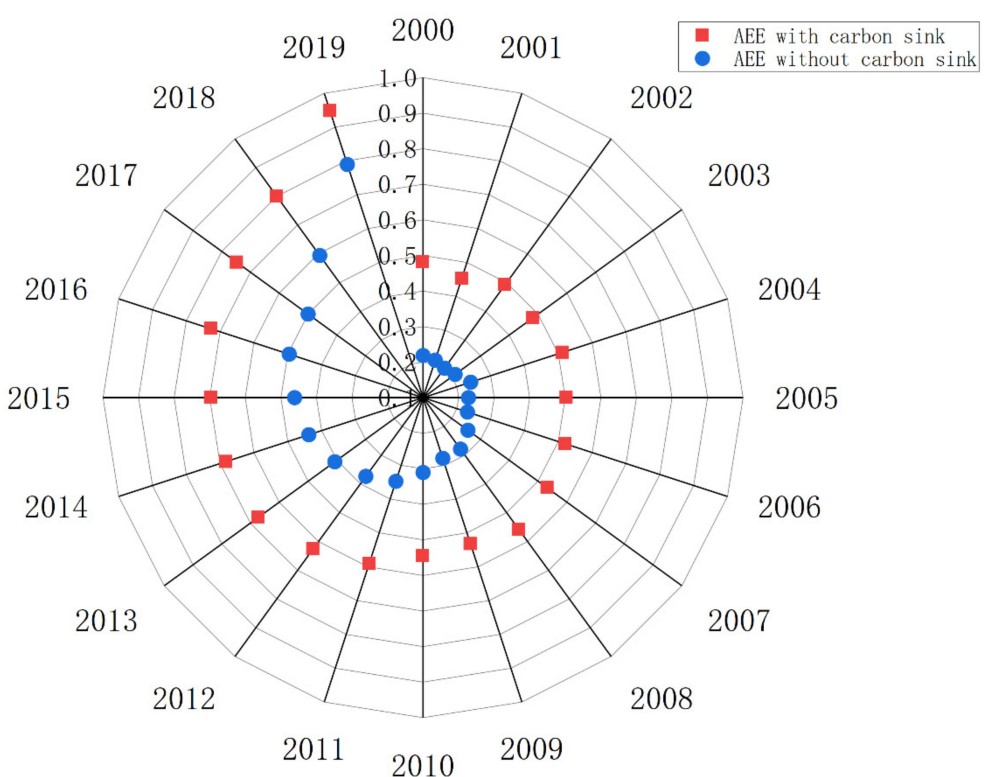

**Figure 1.** China's AEE with and without agricultural carbon sinks for the period 2000–2019.

**Table 3.** AEE ranking of 31 provinces in China with and without agricultural carbon sinks.

| Province | Without Carbon Sink | Rank | With Carbon Sink AEE | Rank | Province | Without Carbon Sink | Rank | With Carbon Sink | Rank |
|---|---|---|---|---|---|---|---|---|---|
| Liaoning | 0.3228 | 16 | 0.6414 | 11 | Guangdong | 0.4357 | 6 | 0.5771 | 16 |
| Jilin | 0.2151 | 26 | 0.8682 | 4 | Guangxi | 0.3007 | 17 | 0.8351 | 5 |
| Heilongjiang | 0.2699 | 19 | 0.6574 | 9 | Hainan | 0.6291 | 2 | 0.8770 | 3 |
| Beijing | 0.5705 | 5 | 0.7553 | 8 | Chongqing | 0.2722 | 18 | 0.5149 | 21 |
| Tianjin | 0.4354 | 7 | 0.6285 | 13 | Sichuan | 0.3625 | 12 | 0.5657 | 18 |
| Hebei | 0.2475 | 23 | 0.4697 | 23 | Guizhou | 0.3400 | 13 | 0.6467 | 10 |
| Shanxi | 0.1580 | 30 | 0.3647 | 30 | Yunnan | 0.1971 | 28 | 0.4488 | 26 |
| Inner Mongolia | 0.2214 | 25 | 0.5113 | 22 | Shaanxi | 0.3962 | 11 | 0.6121 | 14 |
| Shanghai | 0.6177 | 3 | 0.8785 | 2 | Gansu | 0.1300 | 31 | 0.2853 | 31 |
| Jiangsu | 0.3962 | 10 | 0.5879 | 15 | Qinghai | 0.5978 | 4 | 0.6393 | 12 |
| Zhejiang | 0.3346 | 15 | 0.4642 | 24 | Ningxia | 0.3365 | 14 | 0.7683 | 7 |
| Anhui | 0.1692 | 29 | 0.3837 | 29 | Xinjiang | 0.2670 | 20 | 0.8120 | 6 |
| Fujian | 0.4333 | 8 | 0.5204 | 20 | Tibet | 0.9245 | 1 | 1.0025 | 1 |
| Jiangxi | 0.2082 | 27 | 0.4240 | 27 | Northeast region | 0.2693 | 4 | 0.7223 | 1 |
| Shandong | 0.4077 | 9 | 0.5751 | 17 | East region | 0.3500 | 3 | 0.5469 | 4 |
| Henan | 0.2633 | 22 | 0.5652 | 19 | Central region | 0.3556 | 2 | 0.6230 | 3 |
| Hubei | 0.2648 | 21 | 0.4638 | 25 | Western region | 0.3824 | 1 | 0.6296 | 2 |
| Hunan | 0.2397 | 24 | 0.4200 | 28 | Average in China | 0.3537 | | 0.6053 | |

In summary, agricultural carbon sinks are an important factor for accurately assessing AEE. Estimates of AEE may be biased if agricultural carbon sinks are ignored. Therefore, agricultural carbon sinks must be considered when assessing China's AEE. This is consistent with the findings of Liao [51].

### 3.2. Analysis of Time Series Evolution Characteristics of AEE in China

On the national scale, China's AEE showed a clear upward trend from 2000 to 2019. The national AEE increased from 0.48 in 2000 to 0.95 in 2019, a growth rate of 97.92%. However, at the provincial scale, most of China's AEE is at a low level, with only 14 provinces exceeding the national average.

In order to clearly explore the evolution process of the regional differences in AEE over time, the Gaussian kernel function was used to analyze the evolution law based on the Silverman optimal bandwidth (Figure 2). From the perspective of displacement, the Gaussian kernel density curve gradually shifted from left to right from 2000 to 2009, indicating that China's AEE gradually increased, which also confirmed the consistency of the previous research results. In 2000, a single main peak was obvious, and double peaks had the tendency to form but were weak. By 2019, the single peak shifted to the right. In terms of the kurtosis intensity, the main peak shifted to the right between 2000 and 2015, but the change in kurtosis was not obvious. By 2019, the kurtosis had changed from broad to sharp and had sharply increased, reflecting the difference of AEE in provincial areas is relatively small during the period from 2016 to 2019. This may be because China has been paying more attention to the development of agricultural green ecology since 2016, and has successively issued the "Opinions on Accelerating the Construction of Ecological Civilization", "National Agricultural Sustainable Development Experimental Demonstration Zone Construction Plan", "Innovation Institutions and Mechanisms to Promote Agricultural Green Development", "Opinions on Mechanism to Promote Green Agricultural Development", and other relevant policy documents. Through these actions, the country has achieved remarkable results.

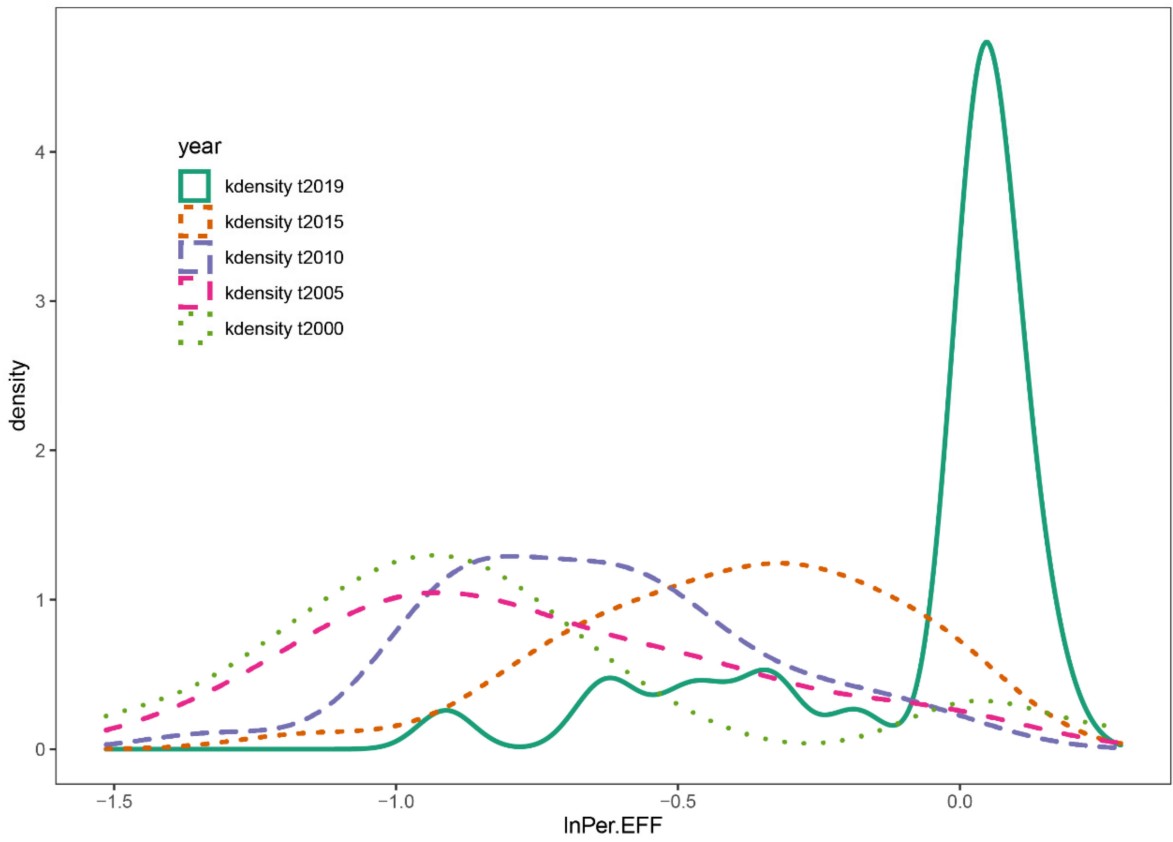

**Figure 2.** Distribution and evolution trend of the AEE core density in representative years.

### 3.3. Spatial Differentiation Characteristics of AEE in China

Based on the 2000, 2005, 2010, 2015, and 2019 data as well as the mean cross-section samples from 2000 to 2019, the spatial distribution map of China's AEE was drawn with the help of ArcGIS software (Figure 3) (The version of the software is ArcGIS Desk top 10.8, by Esri.Inc.). The figure shows that with the development of the economy and society, there were obvious differences in the AEE of Chinese provinces in different years.

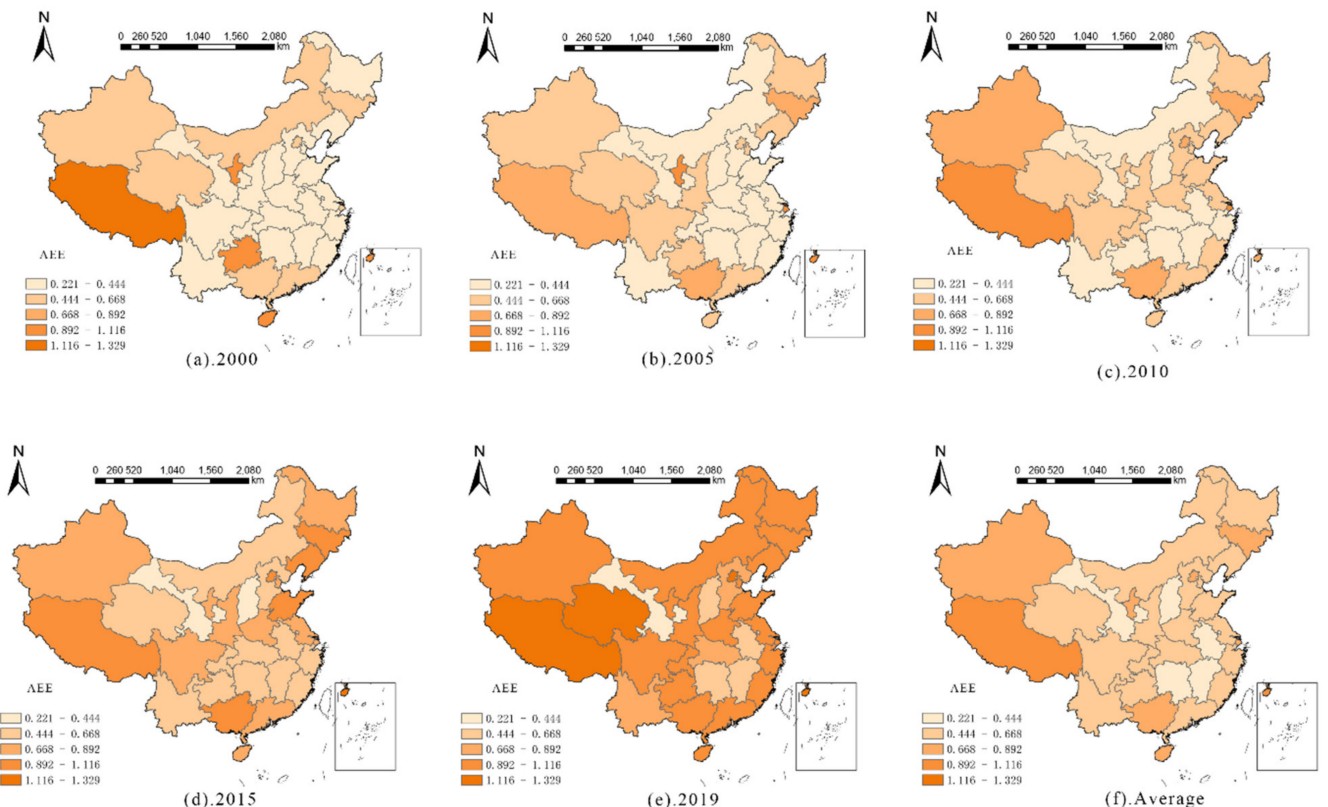

**Figure 3.** Spatial distribution of AEE every five years in China.

The differences in AEE between provinces were also obvious (Figures 3 and 4). Tibet, which is in the western region, had the highest AEE value, which reached 1.10. The next highest AEE values were in Shanghai, Hainan, Jilin, Guangxi, and Xinjiang provinces, with values of 0.88, 0.88, 0.87, 0.84, and 0.81, respectively. The 16 provinces Ningxia, Beijing, Heilongjiang, Guizhou, Liaoning, Qinghai, Tianjin, Shaanxi, Jiangsu, Guangdong, Shandong, Sichuan, Henan, Fujian, Chongqing, and Inner Mongolia had AEE values between 0.51 and 0.77. Hebei, Zhejiang, Hubei, Yunnan, Jiangxi, Hunan, Anhui, Shanxi, and Gansu had low AEE values, all of which were lower than 0.47, with Gansu having the lowest at 0.29 (Table 3).

To further identify regional gaps, the AEE levels of the eastern, central, western, and northeastern regions were compared (Figure 4). The average AEE in the northeastern region was 0.72, the western region was 0.63, the eastern region was 0.62, and the central region was 0.55. As shown in Figure 4, the northeast region had the highest and most consistent AEE, whereas the east and the west had more variable values and the central region had consistently lower values.

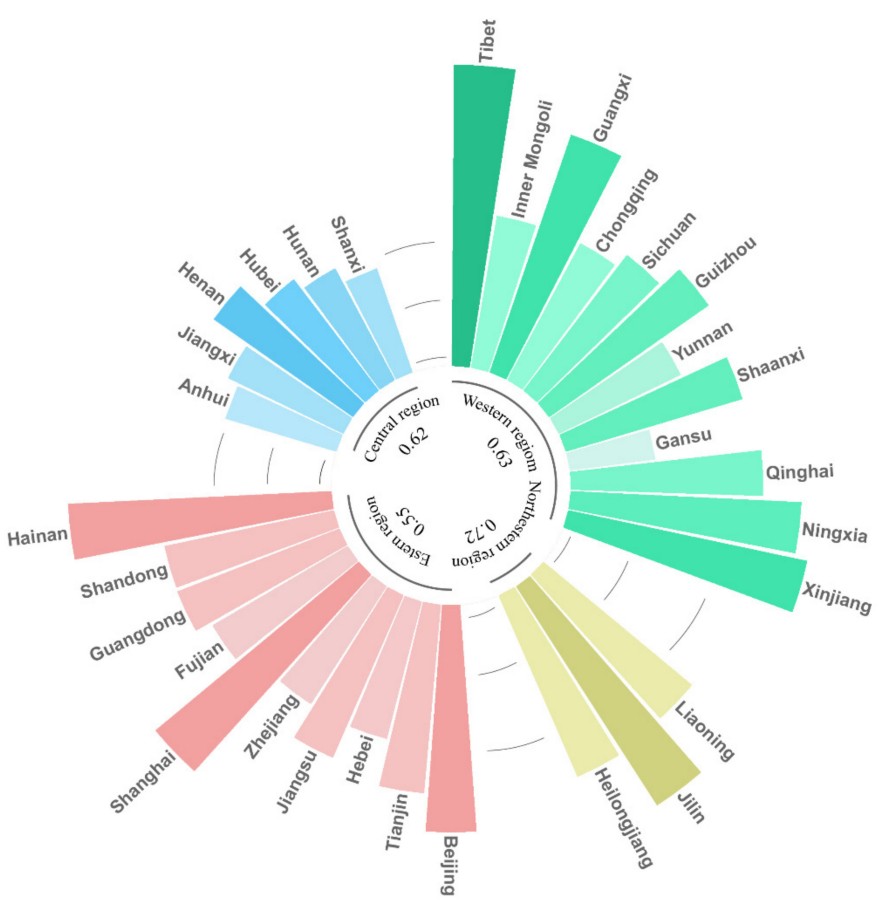

**Figure 4.** Spatial distribution of AEE.

### 3.4. Drivers of Spatial Differentiation of AEE in China

3.4.1. Identification of Driving Factors for the Spatial Differentiation of AEE in China

Using the geographic detector, we obtained and sorted the *q*-values of all factors at five time points (2000, 2005, 2010, 2015, and 2019) and the entire study period (Table 4).

**Table 4.** Factor detection results of the spatial differentiation of AEE in China.

| Factor | 2000 | | 2005 | | 2010 | | 2015 | | 2019 | | Total | |
|---|---|---|---|---|---|---|---|---|---|---|---|---|
| | *q* | Rank | *q* | Rank | *q* | Rank | *q* | Rank | *q* | Rank | *q* | Rank |
| Per capita arable land | 0.24 | 9 | 0.24 | 7 | 0.32 | 4 | 0.15 | 11 | 0.38 | 2 | 0.16 | 11 |
| Agricultural disaster rate | 0.32 | 5 | 0.48 | 1 | 0.39 | 1 | 0.35 | 2 | 0.43 | 1 | 0.22 | 8 |
| Precipitation | 0.35 | 4 | 0.37 | 3 | 0.16 | 10 | 0.31 | 5 | 0.27 | 7 | 0.16 | 10 |
| Agricultural economic level | 0.17 | 10 | 0.19 | 9 | 0.25 | 6 | 0.43 | 1 | 0.23 | 8 | 0.35 | 4 |
| Industrial structure | 0.27 | 7 | 0.18 | 10 | 0.38 | 2 | 0.18 | 10 | 0.34 | 4 | 0.27 | 7 |
| Degree of agricultural mechanization | 0.17 | 11 | 0.25 | 6 | 0.14 | 11 | 0.23 | 9 | 0.30 | 6 | 0.21 | 9 |
| Urbanization level | 0.31 | 6 | 0.28 | 5 | 0.23 | 7 | 0.35 | 3 | 0.33 | 5 | 0.32 | 6 |
| Level of industrialization | 0.75 | 1 | 0.40 | 2 | 0.22 | 8 | 0.28 | 6 | 0.35 | 3 | 0.51 | 1 |
| urban–rural gap | 0.36 | 3 | 0.18 | 11 | 0.19 | 9 | 0.33 | 4 | 0.22 | 9 | 0.40 | 3 |
| Years of education per capita in rural areas | 0.39 | 2 | 0.24 | 8 | 0.33 | 3 | 0.25 | 7 | 0.19 | 11 | 0.34 | 5 |
| The level of financial support for agriculture | 0.26 | 8 | 0.32 | 4 | 0.30 | 5 | 0.25 | 8 | 0.21 | 10 | 0.44 | 2 |

Table 4 shows that each explanatory variable had a different explanatory power for the spatial differentiation of AEE at different time points. From the perspective of the entire study period, the top five factors that affected the differentiation of AEE in China were the level of industrialization ($q = 0.51$), the level of fiscal support for agriculture ($q = 0.44$), the urban–rural gap ($q = 0.40$), the agricultural economic level ($q = 0.35$), and the years of education per capita in rural areas ($q = 0.34$). These five factors had a strong explanatory power and had the greatest impact on the spatial differentiation of AEE. The explanatory power of the agricultural disaster rate, urbanization level, industrial structure, per capita cultivated land area, and agricultural mechanization degree to the spatial differentiation of AEE was relatively general, with a $q$ value between 0.16 and 0.32. The factor with the weakest explanation ability was the per capita cultivated land area, with a $q$ value of only 0.16.

3.4.2. Interaction Identification of the Spatial Differentiation of AEE

In order to explore the AEE explanatory power when different driving factors interact, the interaction of each factor was selected to analyze the interaction mechanism affecting the spatial differentiation of AEE. The average value of the whole study period was used as the sample for analysis, and the results showed the interaction between the factors is close; in addition to mutual independence, both antagonistic and synergistic effects occurred (Figure 5).

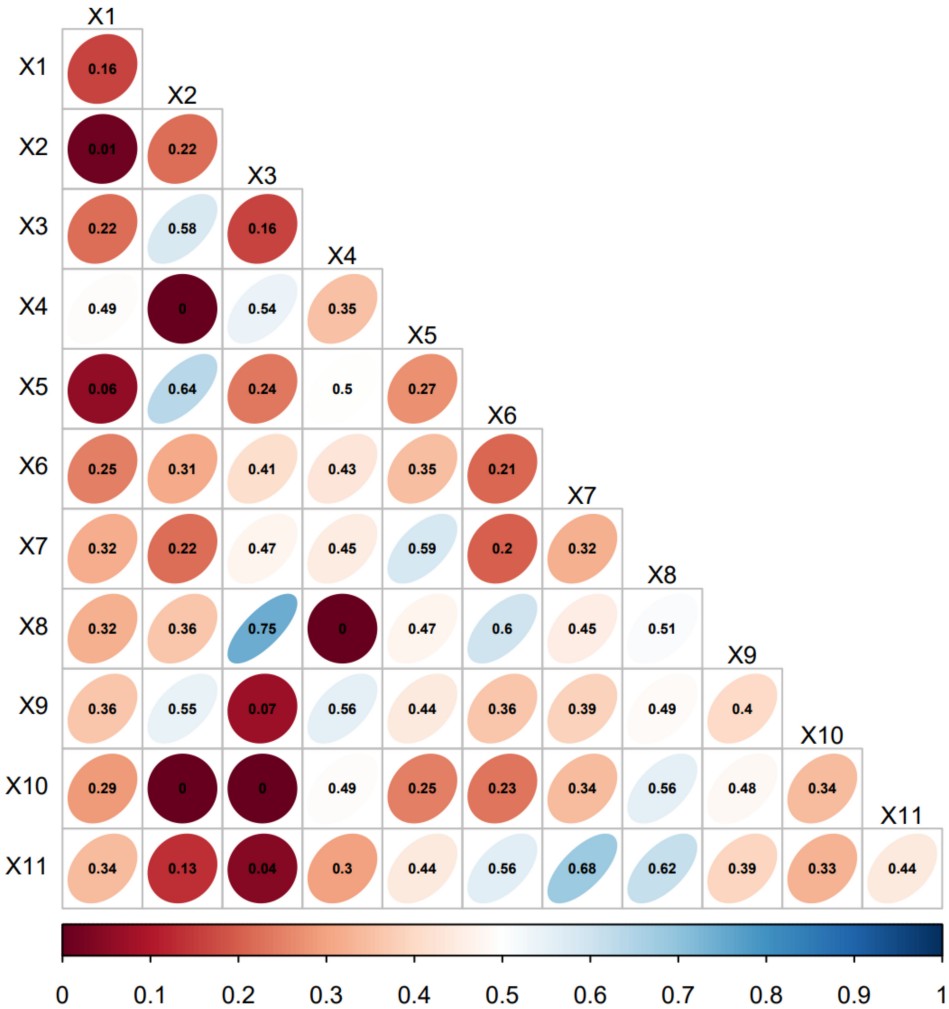

**Figure 5.** Factor interaction results.

The interactions of the agricultural disaster rate with precipitation and the agricultural industrial structure showed nonlinear enhance, as did the interactions of precipitation with the agricultural economic level, agricultural mechanization level, and industrialization level. The agricultural industrial structure and urbanization level also exhibited nonlinear enhance when interacting. The interactions of the per capita arable land area with precipitation, the agricultural economic level, agricultural industrial structure, and agricultural mechanization level showed bivariate enhance. The interactions of the agricultural disaster rate with the agricultural mechanization level and urban–rural gap, as well as the interactions of precipitation with the urbanization level and years of education per capita in rural areas showed bivariate enhance. The interactions of the agricultural economic level with the agricultural industrial structure, agricultural mechanization level, urbanization level, and urban–rural gap, showed bivariate enhance. The interactions of the agricultural industrial structure with the industrialization level, urban–rural gap, years of education per capita in rural areas, the level of financial support to agriculture, urbanization level, and urban–rural gap showed bivariate enhance. The interactions of the agricultural mechanization level with the industrialization level and the level of financial support to agriculture showed bivariate enhance. The interactions of the urbanization level with years of education per capita in rural areas and the level of financial support to agriculture showed bivariate enhance. The interactions of the industrialization level with years of education per capita in rural areas and the level of financial support to agriculture showed bivariate enhance. The interactions of the urban–rural gap with years of education per capita in rural areas showed bivariate enhance. These results show that most of the dominant factors were synergistic when they interacted, though some other factors were antagonistic.

## 4. Discussion

### 4.1. AEE Considering Carbon Sinks

This study analyzed the spatiotemporal differentiation and driving factors of AEE in China. The results of the study found that there is still much room for improvement in China's AEE. In future agricultural production practices, more attention should be paid to green ecological development while pursuing economic benefits [52].

Additionally, there were certain differences in the research results of scholars. For instance, some scholars found that the spatial order of AEE in China is eastern > western > central > northeastern [53,54], while other scholars found that the province with the highest ecological efficiency is Qinghai [55]. This study attributes these differences to two reasons. First, previous studies did not include Tibet, which is an ecological civilization highland, in the research scope. Second, an evaluation index system that includes agricultural ecological welfare was constructed in this study and it considers agricultural carbon sinks as part of the expected output. After adding the agro-ecological welfare factor, the AEE of some eastern coastal provinces decreased, while the AEE of some western and northeastern provinces increased [56]. The AEE frontiers in the northeast and western regions with low degrees of development and high ecological values shifted as a whole, and the overall AEE increased.

### 4.2. Drivers of Spatial Differentiation of AEE

The leading driving factors of AEE differed at different time points. The level of industrialization, per capita years of education in rural areas, overall explanatory power of the urban–rural gap, and the level of financial support for agriculture declined over the study period. This may be because the development of industrialization is conducive to the development of petroleum-based agriculture [33,57]. With the improvement of China's level of industrialization in the 21st century, the impact of the industrialization level on the differentiation of AEE is also weakening. The $q$-value ranking of the per capita years of education in rural areas dropped from 2nd in 2000 to 11th in 2019. This may be the reason why China has greatly promoted the balanced development of education in recent years, the level of education in rural areas has been greatly improved [58]. The impact of the education level

of farmers on the spatial differentiation of AEE is weakening; the *q*-value ranking of the urban–rural gap dropped from 3rd in 2000 to 9th in 2019. This may be because, with the acceleration of China's urbanization process and poverty alleviation in rural areas, the overall victory of the agricultural economy has greatly increased the income of rural residents [59], and the narrowing of the urban–rural gap has also weakened the differentiation of AEE. With the acceleration of urbanization, traditional agriculture-based rural society is rapidly transforming into a modern urban society dominated by non-agricultural industries, and the agricultural population is decreasing [60]; these changes are resulting in the decrease in the differentiation of AEE. The level of financial support to agriculture dropped from 8th place in 2000 to 10th place in 2019. This may be because the rural revitalization strategy implemented by China in 2017 led to provincial governments strongly supporting the development of the agricultural industry and providing considerable political and financial support. In particular, the agricultural development level of poverty-stricken areas has greatly improved, such that the spatial differentiation of provinces is weakening [61,62].

The above results show that there are many factors affecting the spatial differentiation of China's AEE, and the influence of the driving factors significantly differs depending on the stage of social development. Therefore, agricultural development must keep pace with economic and social development, and it must consider the dominant factors to improve their influence on the spatial differentiation of AEE. For this reason, each region should formulate corresponding development strategies based on its own agro-ecological development status, while simultaneously paying attention to inter-regional coordination and cooperation, in order to achieve resource conservation as well as environmentally friendly and high-quality sustainable development [63,64].

*4.3. Measurement Method*

In this study, an index system for measuring AEE was constructed under the premise of considering agricultural carbon sinks, and the super-efficiency SBM model was used to bring the measurement results closer to the reality of agricultural production [65]. Although various factors are considered, ecological efficiency focuses on the coordinated development of economy and environment. The environmental problems caused by agricultural production are not only the production of greenhouse gases, but also cause certain agricultural non-point source pollution [66,67]. Agricultural non-point source pollution is not easy to quantify. Due to the difficulty of data acquisition, the index system constructed in this study does not account for agricultural non-point source pollution. Subsequently, quantitative analysis of agricultural non-point source pollution can be carried out from the residues caused by the use of pesticides, fertilizers, and agricultural films [68]. Furthermore, the spatial differentiation of AEE is affected by many factors. It is crucial to understand the dominant factors of spatial differentiation. Quantitatively solving the spatial differentiation of AEE is also a problem that needs attention. Quantitative analysis of the regional differentiation of the driving factors of AEE through geographic detection can not only quantify the impact of a single factor, but also quantify the interaction between the two factors; thus, this method can effectively solve the qualitative problem of AEE in spatial stratification.

**5. Conclusions**

Based on the perspective of agricultural carbon sinks, this study constructed a measurement index system of China's agricultural ecological efficiency and measured the spatial and temporal evolution, spatial differentiation, and driving factors of regional AEE for 31 provinces in China. The research conclusions were as follows: China's AEE may be underestimated when the agricultural carbon sink is not considered. Using the agricultural carbon sink as the expected output of the measurement will bring the measured AEE closer to reality. China's AEE showed an overall upward trend from 2000 to 2019. However, the overall level of China's AEE was not high, and there is ample room for improvement. The AEE of the four major economic zones was 0.72 in the northeast,

0.63 in the west, 0.62 in the center, and 0.55 in the east. There was also a big difference between provinces. Therefore, the development of regional AEE was unbalanced, and the phenomenon of regional differentiation was obvious. The spatial differentiation of China's AEE was affected by multiple factors, among which the industrialization level, urban–rural gap, agricultural economic level, agricultural disaster rate, and urbanization level were the dominant factors, with AEE varying in certain years. The influence of the agricultural disaster rate, agricultural economic level, and urban–rural gap on the spatial differentiation of AEE gradually increased over time. At the same time, the spatial differentiation of AEE in China was significantly affected by pairs of factors in both antagonistic and synergistic ways. Therefore, based on our own resource endowments, we should strengthen financial policy support, promote the improvement of agricultural ecological efficiency according to the time and place, realize the minimum input of agricultural resource elements and maximize the agricultural economic benefits; at the same time, we should reduce agricultural greenhouse gas emissions and effectively promote high-quality and sustainable agricultural development.

However, this study still contains some shortcomings. For example, the exploration of the factors driving the spatial differentiation of AEE only considers the interaction between two factors and does not explore the interaction between three or more factors. Furthermore, in addition to the visible cost of agricultural production, various natural and socio-economic factors have an impact on the loss of AEE. Future research should conduct an in-depth study of the impacts of natural factors, related policies, agricultural labor, changes in agricultural technology, and other factors on AEE.

**Author Contributions:** Conceptualization, S.L. and Y.F.; methodology, S.L.; software, S.L.; validation, Z.Z., Z.D., D.W. and J.D.; formal analysis, S.L.; investigation, S.L.; writing—original draft preparation, S.L.; writing—review and editing, S.L.; funding acquisition, Y.F. All authors have read and agreed to the published version of the manuscript.

**Funding:** This research was funded by the 2022 Provincial Grain Special Project of Shaanxi Province (Shaanxi Grain Reserve Safety Early Warning and Emergency Management Strategy Research), the 2022 Shaanxi Provincial Association for Science and Technology Decision-making Consulting Project (Research on the Industrial Model of Comprehensive Land Remediation in the Background of Rural Revitalization), and the Shaanxi Provincial Forestry Science and Technology Innovation Program Special Project (grant number SXLK20200102).

**Institutional Review Board Statement:** Not applicable.

**Informed Consent Statement:** Not applicable.

**Data Availability Statement:** The data that support the findings of this study are available from the corresponding author upon reasonable request.

**Conflicts of Interest:** The authors declare no conflict of interest.

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
