# Peer review of "Temporal and Spatial Differentiation and Driving Factors of China’s Agricultural Eco-Efficiency Considering Agricultural Carbon Sinks"

_agriculture, doi:10.3390/agriculture12101726_

Round 1
Reviewer 1 Report
The novelty of the work is the application of the eco-efficiency indicator, implementing three different tools for a better methodological approach: Super-SBM model with unexpected output methodology, Kernel density function and Geodetector Model. The aim consists in measuring the level of regional agricultural sustainability in Chinese regions. In contrast to the other research works, the current study, containing the analysis conducted over 20 years, addresses two new points: first, Tibet is considered in the sample of 31 Chinese regions; second, the evaluation index system constructed in this study also takes into account agricultural ecological well-being by considering agricultural carbon sinks as part of the expected output.
The title is explanatory of the work and the chosen journal reflects the topics covered. The abstract excellently anticipates what the paper is trying to clarify, emphasising sample size, some results and implications. The objective of the research work is clear and made known in the introduction to the article. The language used and topics addressed appear to be exactly aligned with the publisher to whom the work was submitted.
Subsequently, the sections are expanded for some more specific comments.
Introduction
The introduction explains the context and intent of the research work. However, in order to increase the attractiveness of the work, it is recommended to provide insight in terms of why China in particular was examined and why it may be so interesting to analyse this country (part of this can also be deduced from the conclusions). Consequently, it also creates more appeal for reading the Chinese case study.
Material and Methods
The source from which the variables were extracted is included in section 2.1, but it is not clear and easy to visualise. It is therefore suggested to make this paragraph more discursive or opt for inserting the data source in one of the summary tables or to create one specifically, including the units of measurement. It is considered to be essential to include the following work to enhance this section:
Frittelli, M., Madzvamuse, A., Sgura, I., & Venkataraman, C. (2018). Numerical preservation of velocity induced invariant regions for reaction–diffusion systems on evolving surfaces. Journal of Scientific Computing, 77(2), 971-1000.
Frittelli, M., Madzvamuse, A., & Sgura, I. (2021). Bulk-surface virtual element method for systems of PDEs in two-space dimensions. Numerische Mathematik, 147(2), 305-348.
Lacitignola, D., Frittelli, M., Cusimano, V., & De Gaetano, A. (2022). Pattern formation on a growing oblate spheroid. An application to adult sea urchin development. Journal of Computational Dynamics, 9(2), 185.
The variables chosen are suitable for the type of study being conducted, as well as for the research questions proposed in the introductory section. Moreover, the summary tables provide a clear view of the variables under study, whereby, as marked above, it is advisable to add a column dedicated to the source of the variables.
Results
The graphs and tables are significant for the phenomena to be illustrated and the readings taken on the indices that have been taken into account are approved.
Do the four colours used serve to distinguish the four areas in Figure 4? If the answer is yes, one might even consider going beyond the use of four different colours, as the four groups of regions are already spaced out. Furthermore, there are no comments in the body of the text that recall the colours of the figure. So, one could use the same colour as in Figure 3 (orange), with different gradations depending on the value examined for each region.
Otherwise, the results of the study are well-explained and comprehensively illustrated.
Discussion
The discussion is interesting and points out the novelty aspects compared to other works. It also concatenates the reasons behind the results obtained. Section 4.3 focuses on the strengths and weaknesses of the work, in a complete overview.
Conclusion
Is Point 4 the answer to the last research questions posed in the introduction? Is it correct to think that the introduction session and the conclusion session are interconnected through these bullet points? If the answer is correct, it is suggested to clarify better this link in the conclusion. The implications are dutiful and essential. The inclusion of limitations in the work is highly appreciated.
Reviewer 2 Report
In general, the manuscript is well designed and nicely presented and can be accepted after major revision. The following points need to be addressed before acceptance.
1. The abstract must be restructured in view of the reflection of core findings, and concluding lines.
2. Introduction must be reframed, hypothesis and objective of the study must be clear.
3. Methodology is clear
4. Results and discussion is ok but needs to be supported with published reports from the same line.
5 Conclusion is too lengthy and must be reduced to one paragraph. The conclusion must be client oriented.
6. Overall article length of the article must be reduced.
7. References should be cross-checked. Please follow the pattern and style of the journal.
Round 2
Reviewer 2 Report
The authors addressed all my suggestions meticulously now the MNS can be accepted in its current form.